# PIWI-Interacting RNAs: A Pivotal Regulator in Neurological Development and Disease

**DOI:** 10.3390/genes15060653

**Published:** 2024-05-21

**Authors:** Xian Pan, Wang Dai, Zhenzhen Wang, Siqi Li, Tao Sun, Nan Miao

**Affiliations:** Center for Precision Medicine, School of Medicine and School of Biomedical Sciences, Huaqiao University, Xiamen 361021, China; 22013071022@stu.hqu.edu.cn (X.P.); 2026312005@stu.hqu.edu.cn (W.D.); 23014071016@stu.hqu.edu.cn (Z.W.); 23013071016@stu.hqu.edu.cn (S.L.); taosun@hqu.edu.cn (T.S.)

**Keywords:** piRNAs, PIWI protein, central neural system disorder, biogenesis

## Abstract

PIWI-interacting RNAs (piRNAs), a class of small non-coding RNAs (sncRNAs) with 24–32 nucleotides (nt), were initially identified in the reproductive system. Unlike microRNAs (miRNAs) or small interfering RNAs (siRNAs), piRNAs normally guide P-element-induced wimpy testis protein (PIWI) families to slice extensively complementary transposon transcripts without the seed pairing. Numerous studies have shown that piRNAs are abundantly expressed in the brain, and many of them are aberrantly regulated in central neural system (CNS) disorders. However, the role of piRNAs in the related developmental and pathological processes is unclear. The elucidation of piRNAs/PIWI would greatly improve the understanding of CNS development and ultimately lead to novel strategies to treat neural diseases. In this review, we summarized the relevant structure, properties, and databases of piRNAs and their functional roles in neural development and degenerative disorders. We hope that future studies of these piRNAs will facilitate the development of RNA-based therapeutics for CNS disorders.

## 1. Introduction

PiRNAs, the largest subclass of sncRNAs in the PIWI family [1], were initially identified in the reproductive system [2,3]. The mature piRNAs are generated from either an initial transcript originating from a piRNA cluster or the conversion of double-stranded RNA precursors by RNase type-III enzymes [4,5,6]. PiRNAs bind and guide PIWI proteins to target specific mRNAs through sequence complementarity, and play important roles in gene regulation, transposon element (TE) repression, and antiviral defense [7,8]. Compared to other small RNAs, piRNAs utilize a more adaptable approach to silencing target mRNA, albeit necessitating longer complementary sequences [7,9].

In germ cells, piRNAs normally maintain gene integrity by silencing TEs [10,11,12,13]. PiRNAs/PIWI are also essential for regulating spermatogenesis/oogenesis [10,14], such as PIWI1/2/3 deficiency causing male mouse sterility by promoting aberrant transposon and transcriptome activity [15,16,17,18].

An increasing number of studies have highlighted the potential roles of the piRNAs-genes/lncRNAs complex in regulating neuronal cell differentiation, migration, and development [19,20]. PiRNAs are also involved in various neurological processes, including cognition, learning, and memory-related behaviors [21,22,23]. What is more, dysregulation of the piRNAs/PIWI pathway is linked to neurodevelopmental and degenerative diseases [24,25,26,27]. For example, in two Parkinson’s disease *Caenorhabditis elegans* models, *21ur-10824*, *21ur-11898*, and *21ur-13215* are significantly up-regulated in both α-syn(A53T)^Tg^ and Aβ^Tg^ and α-syn(A53T)^Tg^ transgenic strains [28].

PiRNAs have been increasingly recognized for their role in CNS development, including neural stem cell (NSC) differentiation and neurocognitive recovery, as well as in neurodevelopmental and neurodegenerative activities [20,24,25,27]. To this end, this review aims to summarize the progress in elucidating the structure, properties, and regulatory roles of piRNAs, particularly in CNS development. It also describes their potential applications in neurodevelopmental and degenerative diseases, both in human and animal models, thereby providing valuable insights for future research and clinical practice in neural disorders.

## 2. The Basic Information of piRNAs

To gain a better understanding of the piRNAs/PIWI complex, we will review the biogenesis, classification, distribution, and relevant databases of piRNAs in this section.

### 2.1. piRNA Biogenesis

PiRNA biogenesis is a multistep process starting with transcription and the early processing of precursor RNA in the nucleus, export, and precursor cleavage, followed by processing to mature piRNAs and loading into PIWI proteins [29]. It includes two steps: the primary loop and the ping-pong loop (Figure 1) [30,31]. 

In the primary loop, piRNA precursors originate from the piRNA cluster loci [29]. In the nucleus, the piRNA precursors are exposed to the Yb body and cleaved by the endonuclease Zuc, generating intermediate piRNAs, with the first nucleotide being uridine at the 5′ end [32,33]. These intermediate piRNAs further undergo trimming at their 3′ ends, eventually producing mature initiator piRNAs [34,35]. Finally, these mature initiator piRNAs are transported into the nucleus [32,36]. Thus, the primary loop initiates the downstream response of the ping-pong loop.

A secondary pathway, also called a ping-pong loop, involves the processing of pre-piRNAs initiated by the slicer activity of cytoplasmic PIWI proteins Aub or Ago3, loaded with complementary piRNAs [4,37]. To produce the responder pre-piRNAs in the ping-pong cycle, the initiator piRNAs bind PIWI proteins and are cleaved in a Zuc-dependent or independent manner [38,39]. To expand the diversity of piRNAs, the piRNA/AGO3 complex cuts the target RNA [40]. Taken together, the ping-pong pathway functions as an amplification cycle to increase the abundance and diversity of piRNAs.

### 2.2. The Structure, Classification, and Distribution of the piRNA/PIWI Complex

#### 2.2.1. Structure of the piRNA/PIWI Complex

As a member of the Argonaute protein family, the structures of PIWI proteins are composed of PAZ in the N-terminal, RNase H-like, and Piwi/Argonaute/Zwille domain in the C-terminal [41]. The N-terminal PAZ can be attributed to PIWI protein–small RNA complex formation [42]. In the C-terminal, the RNase H-like domain shows RNA endonuclease activity in RNA-DNA hybrid molecules [43], and the Piwi/Argonaute/Zwille domain participates in the RNA interference process by binding piRNAs [44]. The PIWI proteins are highly conserved across multiple species [45,46,47], such as PIWI, Aub, and Ago3 in the fruit fly [48], MIWI/MWI2 (PIWIL1/4) and MILI (PIWIL2) in the mouse [49] and HIWI (PIWIL1), HILI (PIWIL2), HIWI3 (PIWIL3), and HIWI2 (PIWIL4) in humans [50]. 

Mature piRNAs are around 24–32 nt, with a preference for uridine at the 5′ end and 2′O-methylation at the 3′ end [35,45,51]. Compared to miRNAs, piRNAs usually lack sequence conservation and Dicer biogenesis [52]. Moreover, piRNAs are more selective than miRNAs when identifying a target through a weak seed region [53]. To ensure valid silencing TE activity, piRNAs closely monitor the sequence pairing through the seed gate [53]. Unlike other sncRNAs, piRNAs have been found only in animals and appear across different species [54].

#### 2.2.2. PiRNAs in Different Genomic Areas and Cell Types

In most mammals, mature piRNAs are mainly divided into three major categories: piRNAs in TE, intergenic piRNAs, and 3′-UTR [55,56,57,58,59]. As the major class, TE piRNAs typically target the transcription and translation of TEs by RNAi in *Drosophila* and *Danio rerio* [60,61]. Intergenic piRNAs are mainly found in adult mammalian testes, such as in mice, rats, and humans [62,63,64]. In mice and chickens, the precursors of 3′UTR piRNAs are guided post-termination by 80S ribosomes, which harbor embedded TEs and produce piRNAs that cleave these TEs [65]. However, over 80% of piRNAs (including TE, 3′UTR, and intergenic piRNAs) lack obvious targets in mammals [65,66]. Overall, while piRNAs are associated with diverse genomic functions, their specific produced source in mammals still requires further exploration. 

PiRNAs and PIWI are also abundant in multiple somatic cells, such as in neuronal, liver, exosome, and cancer tissue cells [21,29,34,67]. The aberrant expression of ectopic piRNAs has been identified across a range of human diseases. For example, in the CNS, PIWI and piRNAs have been linked to neural differentiation [68,69], dendritic complexity [21,22], and neural crest specification [70]. These findings highlight the broad applicability of piRNAs as potential biomarkers and therapeutic targets in various diseases, especially in CNS disorders. Further research underlying piRNA dysregulation in different neural cell types is necessary to fully elucidate their therapeutic potential.

### 2.3. The Databases of piRNAs

Due to the lack of conserved sequences and structural features, distinguishing piRNAs from miRNAs and siRNAs is a challenge [71]. Numerous piRNA clusters have been annotated, and the relevant databases have been generated [7]. Here, we review the public and specific databases of piRNA research in Table 1.

Several public databases provide comprehensive information on piRNA analysis [73,75]. NONCODE and deepBase 3.0 provide basic information about the expression profile, mapping, potential function, and interaction function of piRNAs [72,73,74,75]. However, these databases cover only a limited number of species and rarely include the functions and target analyses of piRNAs.

PiRNAbank is the first piRNA database, including annotated information, a sequence homology-based search, clusters, corresponding genes, and repeat elements [77]. As the largest database, piRBase covers approximately 181 million sequences and includes 440 datasets from 44 species for piRNAs [81,82]. What is more, version 3.0 integrates piRNA annotation in all aspects, including the piRNA gold standard of sets, clusters, variants, splicing junction, and expression [76]. Nevertheless, piRBase v3.0 has collected data on diseases and visualizes the piRNAs–targets network [76]. Other databases, such as PiRNAclusters, piRTarBase, and piRNA-IPdb have also been used for cluster identification and target prediction [79,80]. 

These databases provide an excellent piRNA research platform for piRNA analysis in terms of functional analysis and multiple pathological processes [83,84,85]. Therefore, further data updating can be on improving naming conventions, experimental verification of the specific piRNA in diseases, data integration, annotation information, and the target prediction standard. 

## 3. PiRNAs/PIWI in Neural Differentiation and Neurocognitive Behaviors

In the past two decades, piRNAs have been characterized in neuronal cells and are shown to have indispensable roles in brain function and neuronal diseases [22,24,86]. However, the precise mechanism remains unclear. To further enhance our understanding, we summarize the multiple functions of piRNAs/PIWI in neuronal differentiation, the blood–brain barrier (BBB) and neurocognitive behaviors (learning, memory, and anxiety), food intake, and neuron injury.

### 3.1. Neuronal Differentiation

#### 3.1.1. PIWI Proteins

In PIWI families, PIWIL1 and PIWIL2 have been mostly proposed to control memory and anxiety [87,88]. The first evidence that the PIWI protein exists in the brain was found in the *Aplysia* hippocampus [21]. The author suggested that down-regulated MILI (PIWIL2) improves memory-relative behavior by demethylating CREB2 [21,22]. Similarly, in a rat cerebral ischemic injury model, the attenuation of PIWIL2 also increases CREB2 by promoting DNM3A methylation, ultimately preventing cerebral ischemic injury [89]. Recent research also showed that PIWIL2 is essential for proper neurogenesis in the postnatal mouse hippocampus. The depletion of PIWIL2 and piRNAs in the adult hippocampus impairs adult neural progenitor cells’ differentiation toward a neural fate, induces senescence, and generates reactive glia [88]. Overall, one of the possible neuronal regulatory pathways involving PIWIL2 is the passive demethylation of DNM3A.

In the E14.5 cerebral cortex of rats, PIWIL1 silencing leads to neuronal polarization and migration by modulating the microtubule-associated proteins (MAPs, MAP2, and MAP2B) [87]. In the chicken neural crest region, Chiwi (PIWIL1) is markedly higher at early neurulation stages, while Chili (PIWIL2) shows low transcript levels at all stages [70]. The loss of PIWIL1 impedes neural crest specification and emigration by targeting ERNI in the Sox2 pathway [70]. What is more, knockdown PIWIL1 inhibits food intake by targeting NPF1, a type of neuropeptide in the brain correlated with appetite, dietary behaviors, and energy metabolism [90].

Along with PIWIL1 and PIWIL2, PIWIL4 is essential for human embryonic mediated neuronal differentiation [68]. The knockdown of PIWIL4 suppresses the partial recovery of embryonic stem cell markers (MAP2 and TUB33) in NT2 cells. In contrast, overexpressed PIWIL4 decreases OCT4A and NANOG expression [68]. To sum up, PIWI proteins can act as modulators in multiple neural differentiation processes, influencing various pathways such as methylation, MAPs, and Sox2. Thus, more sophisticated studies are needed in the future to further address the potential redundant and synergistic functions of different PIWI family members in cortical development.

#### 3.1.2. PiRNAs

In terms of the CNS, piRNAs were initially identified in E15.5 cultures and the 6-week mouse hippocampus [69]. Some well-designed reviews have summarized the function of piRNAs in neural heterogeneity, neurogenesis, and neural plasticity [91,92]. In this part, we review their regulatory roles in neurogenesis.

Among multiple piRNAs tested in the mouse hippocampus, five uniquely expressed piRNAs (*DQ541777*, *705026*, *719597*, *689086*, and *540285*) were co-immunoprecipitated with MIWI [69]. Lee et al. found that *DQ541777* extended throughout the dendrites in a punctate pattern in both E15 cultured mouse and E18 rat hippocampal neurons. What is more, the inhibition of *DQ541777* activity leads to a statistically significant decrease in dendrite spine area, suggesting its role as a modulator of dendritic spine development [69]. During the retinoic acid (RA)-induced neuronal differentiation of NT2 cells, piRNAs (*DQ582359* and *DQ596268*) are up-regulated [68]. Overall, neuronal piRNAs are abundant during the critical period of neural differentiation and respond to changes in PIWI protein levels.

In adult neural stem cells (aNSCs), piRNAs are focused on synaptic plasticity and memory control in the hippocampus [88,93]. The hippocampus shows the highest number of unique piRNA transcripts (5494 piRNAs) in the adult brain [94]. Recently, it was demonstrated that the piRNA pathway has a crucial role in maintaining the fate of hippocampal NSCs [88]. Researchers identified 298 piRNA clusters that are differentially expressed in mouse undifferentiated aNPCs, among which *piR-cluster 1* is homologous to the human *piR-61648* [88]. Moreover, inhibiting the piRNA pathway in adult NPCs leads to senescence-associated inflammation and an increase in reactive astrogliogenesis [88]. However, these mechanisms of piRNAs in translation and epigenetics have been poorly investigated in the context of the CNS.

These results indicate that piRNAs/PIWI function as early and late neuronal marker regulators, enhancing neuronal differentiation. Future research on the piRNAs/PIWI process will contribute to understanding various CNS diseases, including neurodevelopmental and neurodegenerative diseases.

### 3.2. The Blood–Brain Barrier (BBB)

The BBB is a neuroprotective layer that maintains the homeostasis of the central nervous system and provides an appropriate environment in which neurons can execute their functions [95,96]. In addition, down-regulated HIWI2/PIWIL4 protein suppresses two key tight junctions (CLDN1 and TJP1) through the increased phosphorylation of the AKT/GSJ3 signaling pathway in the BBB [97,98]. Increased PIWIL4 leads to the increased infiltration of inflammatory molecules, potentially disrupting the BBB balance [97]. This evidence profoundly supports the novel concept that piRNAs/PIWI-like proteins may have a potential role in governing the BBB by altering the tight junctions and inflammatory factors.

As a potential marker in regulating the BBB balance and in relevant diseases, the role of piRNAs has not yet been investigated. Therefore, further studies are needed to enhance our understanding of how piRNAs regulate the BBB balance.

### 3.3. Neurocognitive Behaviors (Learning, Memory, Anxiety, and Food Intake)

PiRNAs can act as potential biomarkers in neurocognitive behaviors. In this section, we summarize the potential roles of piRNAs in learning, memory, anxiety, and food intake.

#### 3.3.1. Learning and Memory

Learning-related neuronal plasticity and the formation of memory depend on tightly controlled gene and ncRNA patterns [99]. Increasing evidence implicates piRNAs in learning and memory [22,100]. In the 5-HT induction of *Aplysia* CNS, up-regulated *aca-piR-4*/*15* enhances memory-related plasticity by inhibiting DNA methylation levels in the CREB2 promoter [21]. Similarly, in the sensory neurons of *Aplysia*, the *piR-F* in TE facilitates serotonin-dependent CREB2 methylation [100]. These studies reveal the dynamic mechanisms underlying epigenetic regulation in long-term memory formation. To sum up, in *Aplysia* long-term memory formation, piRNAs regulate CREB2 methylation through promotor and TE factors. Future research should focus on the potential role of the piRNAs/PIWI pathway in mammalian memory-like behavior.

#### 3.3.2. Anxiety

Since the 20th century, anxiety has gradually been recognized as a psychiatric disorder affecting normal emotional behavior [101]. The relevant research has suggested that the piRNA/PIWI complex plays a fundamental role in the pathology of anxiety. In the dorsal hippocampus of mice, knocking down PIWIL1/2 enhances the long-term fear memory without affecting anxiety or locomotion [22]. In contrast, selectively knocking down PIWIL2 increases movement velocity and distance, which is indicative of anxiety-like behavior [22]. It is only downregulating PIWIL1 that shows no observable behavioral effect. Other research shows that knocking down PIWIL2 in the dorsal hippocampus of adult mice is sufficient to reproduce hyperactivity [102]. In a PIWIL2 KO mouse model, DNA is preferentially hypomethylated within the long intersoersed nuclear element1 (LINE1) promoter [102]. Moreover, large numbers of LINE1-derived piRNAs have been identified [102]. In these PIWI protein KO models of anxiety, specific piRNAs have not been implicated. The possible mechanism in terms of behavior regulation is that endogenous siRNAs may bind to piRNAs, thereby silencing brain-expressed retrotransposons. Thus, the precise roles of piRNAs in anxiety still need to be elucidated.

#### 3.3.3. Food Intake

Food intake, as a complex physiological process, is regulated by the CNS through neuropeptides and neurotransmitters [103]. In regulating food intake, specific genetic elements such as genes and small non-coding RNAs, including piRNAs, have been highlighted for their potential [103,104,105,106]. In the locust brain, PIWI1 and Ago3 exhibited relatively higher expression levels, but no signals of PIWI2 were detected, suggesting that PIWI1 and Ago3 may have a fundamental function in the CNS [90]. The knockdown of PIWI1 leads to the suppression of body weight in *locust* by decreasing the expression of neuropeptide NPF1, but not after the knockdown of Ago3 [90]. In addition, the intronic *piRs-3-I3* might enhance the RNA splicing of NPF1 by preventing hairpin formation at the branch point sites, indicating that the piRNAs/PIWI complex has a regulatory role in food intake [90]. Few studies have been conducted in this field. Therefore, ongoing research is exploring their exact roles in mammalian models and their therapeutic potential for conditions related to food intake. 

#### 3.3.4. Axon Injury

In the adult mammalian CNS, injured axons fail to regenerate [107]. In addition, piRNAs are associated with axon regeneration following neural injury [52]. 

In the *C. elegans* sensory system, piRNA transcriptions (PRG-1, TOFU-7, and PRDE-1) are involved in suppressing axon regrowth by regulating *parn-1* and *henn-1* in an independent transcription manner [20]. In the adult rat sciatic nerve, *piR-1199* and *piR-5781* were up- and down-regulated, separately [108]. Additionally, *piR-219* and *piR-1200* were initially down-regulated, returned to the baseline levels, and then showed an up-regulation trend [108]. In a cultured sensory neuron, knocking down the MIWI/piRNAs complex increased axon length by 60% and decreased axon retraction [108]. This article highlights the roles of piRNAs/PIWI in regulating the multiple transcription factors involved in sensory regrowth. Interestingly, after peripheral nerve injury in a mouse model, the depletion of MIWI/PIWIL1 enhanced a series of piRNA expressions (*piR-14384*, *-69959*, *-52953*, and *-52274*) [19]. Compared to the healthy group, depleted MIWI/PIWIL1 inhibited the EGR2, a neural transcription factor localized with the Schwan cell marker S100 [19]. Overall, piRNA-MIWI has a critical function in axon regrowth and repair after nerve injury in mouse models. 

Although the piRNAs/PIWI complex has been reported in various neuronal behaviors, most research has concentrated on identifying piRNAs. Therefore, future studies should focus on improving the specific amplification and synthesis pathways of piRNAs in the nervous system.

## 4. piRNAs/PIWI in Neurodevelopmental and Psychiatric Disorders

Multiple genes and ncRNAs lead to neurodevelopmental dysregulation, which gradually becomes a form of language disorders and brain dysfunction [109,110,111,112]. Unexpectedly, piRNAs show potential roles in neurodevelopmental disorders by directly or indirectly regulating gene/lncRNA [9,45,113]. Therefore, we summarize the underlying roles of piRNAs in stroke and psychiatric disorders (Table 2 and Figure 2), which is essential for improving the relevant diagnosis and treatment.

### 4.1. Stroke

Stroke is an acute cerebral vascular event with high mortality and morbidity. Recent studies have demonstrated that dysregulated piRNAs are strongly associated with stroke pathology by regulating the genes, proteins, and relevant signaling pathways [119,120,121].

In an ischemia rat brain, 54 up- and 51 down-regulated piRNAs were detected, with *piR-177729* and *piR-169523* showing the most significant alterations [114]. What is more, stroke-responsive *piR-143106*, *-177729*, *-169523*, and *-70903* were predicted to target retrotransposon (RT) classes, as well as the zinc finger and Kruppel family [114]. These results suggest that abnormal piRNAs may lead to abnormal transposon expression, resulting in pathophysiologic changes after a stroke. In other research using an ischemia tolerance rat model, 5574 piRNAs were found, with *rno-piR-000618*, *-017990*, and *-014971* up-regulated by targeting CREB2 [89]. Overall, stroke-responsive piRNAs restore the plasticity of damaged nerves and improve memory function after transient global cerebral ischemia.

Although mRNA and ncRNA therapies, including those with piRNAs, have great potential for treating neuronal diseases [121,122,123,124], only the identities of piRNAs have been explored so far. Understanding the piRNA/PIWI complex is likely to bring significant breakthroughs in understanding the complex interactions and molecular mechanisms activated after cerebral ischemia. Therefore, future research needs to discover potential piRNA targets and their associated proteins in terms of the pathophysiology of stroke.

### 4.2. Psychiatric Disorders

Psychiatric disorders are highly prevalent and cause an enormous burden of suffering, lost productivity, morbidity, and mortality [125]. Recent results also show the potential role of piRNAs in psychiatric behaviors, including schizophrenia and major depressive and bipolar disorders [126,127]. 

#### 4.2.1. Schizophrenia (SCZ)

One study identified 37 piRNAs in the anterior cingulate cortex (ACC) of SCZ patients, with one piRNA being correlated with antipsychotic medication [127]. However, the authors did not demonstrate a change in level between the SCZ and control groups. In addition, *piR-15451* and *piR-12246* are down-regulated in the anterior cingulate cortex of schizophrenia subjects [127]. Conversely, *piR-18252* has been correlated with antipsychotic medication issues [127]. However, the authors did not mention the changes seen between the schizophrenia and control subjects. Unlike with miRNAs, few studies have reported on correlations between piRNAs and major depression/bipolar disorder. One article mentioned that two circulated piRNAs (*piR-007899* and *piR-019162*) might be related to symptoms following Gulf War Illness (GWI)-relevant exposure in rats [115]. CNS impairment issues, including memory dysregulation and depression, are among the most common symptoms reported in GWI. 

However, only a few piRNAs and their relevant mechanisms have been identified in psychiatric processes. The possible reasons for this include the scarcity of clinical samples and the difficulty of constructing models. Therefore, detecting more specific piRNAs remains a major challenge.

#### 4.2.2. Autism Spectrum Disorder (ASD)

ASD generally manifests in childhood and mainly presents with isolation, emotional dysregulation, and intellectual disability [128]. The pathological mechanism mainly involves multi-gene management [123,129]. The piRNAs/PIWI complex has also been identified as a potential regulator and diagnostic marker in ASD [130,131]. Knockdown PIWIL1 causes the delayed transition of newborn cortical neurons by enhancing MAPs, suggesting a relationship between the PIWI family and autism [87]. Compared to the healthy group, 21 up-regulated (most significant: *hsa*-*piR-1282*) and 16 down-regulated piRNAs (most significant: *hsa-piR-32159*) were identified in ASD samples [131]. Although 22 up-regulated piRNAs (most significant: hsa-piR-22380) and 7 down-regulated piRNAs (most significant: hsa-piR-27623) were found in the severe symptom vs. mild symptoms group, further validation experiments are needed to confirm their potential as diagnostic biomarkers. In addition, piRNAs in the digestive system and gut microbiota have been implicated in ASD [132]. Similarly, in one study of the saliva of children with autism, *hsa-piR-6148*, *hsa-piR-6145*, *hsa-piR-6147*, *hsa-piR-6146*, and *hsa-piR-6144* were down-regulated and were associated with gastrointestinal (GI) disorders [132]. In ASD fecal samples, in one subject with ASD and a neurotypical control, *hsa-piR-6691*, *-6693*, and *-29205* were significantly down-regulated, while *hsa-piR-28269, -32987*, and *-28059* were up-regulated [130]. In the fecal samples of males with ASD, 66 up-regulated and 256 down-regulated piRNAs were identified [130]. In addition, the authors confirmed that piRNAs can be released into the gut lumen and may interact with the micro- and mycobiota, primarily targeting *NACC1*, *SMAD4*, *TNRC6B*, and *TTN* [130]. This not only provides a novel approach to the clinical diagnosis and analysis of ASD but also confirms the potential of piRNA as a biomarker. 

Overall, the available evidence reflects a strong association between dysregulated piRNAs/PIWI and ASD [132,133]. From target prediction, over 50% of the targets of piRNAs may be lncRNAs, which are most abundant in the CNS. Therefore, further research will be focused on possible lncRNA targets and functional networks. 

## 5. PiRNAs/PIWI in Neurodegenerative Disease

Exhibiting neuron degeneration in specific brain areas, neurodegenerative disorders are a class of pathological conditions that cause a lack of motor coordination and cognitive and memory impairment [134,135,136]. Several well-designed reviews have highlighted the dysregulation of the piRNA pathways in Alzheimer’s disease (AD), Parkinson’s disease (PD), and amyotrophic lateral sclerosis (ALS) [24,26,137]. To enhance understanding, we summarize the potential mechanisms of piRNAs/PIWI in both animal models and patients with neurodegenerative diseases (Table 2 and Figure 2).

### 5.1. Alzheimer’s Disease (AD)

AD is the most common neurodegenerative disease, characterized by progressive cognitive and functional decline [138,139]. Due to its complex pathology, a specific treatment strategy remains ambiguous. However, previous studies suggest that ncRNAs regulate the amyloid precursor protein (APP), tau, amyloid-β (Aβ) peptide, inflammation, and cell death in AD [140,141,142,143]. What is more, recent evidence highlights the potential roles of piRNAs in both AD models and patients [138,144].

#### 5.1.1. AD Models

In tau-transgenic *Drosophila*, heterochromatin loss led to aberrant Ago3 expression [117]. When knocking down Ago3/PIWIL1 in tau-transgenic *Drosophila* brains, fewer apoptotic and proliferating cell nuclear antigen (PCNA)-positive cells were found, and the locomotor defect was reduced [27]. In the HASN^539T^ *C. elegans* model, *tdp-1* knock-out reduced the number of piRNAs from 112 to 0 [116]. Among them, *21ur-10488* and *21ur-2781* were predicted to target *gsk-3* and *tpi-1*, which are related to the AD pathology [116]. These results show that the piRNAs/PIWI complex can serve as an ideal biomarker for therapy and diagnosis. However, few mammalian AD models have been reported.

#### 5.1.2. AD Patients

In AD patients, Roy, Sarkar, et al. identified 146 up- and 3 down-regulated piRNAs [145]. Among them, the AD-associated genes, CYCS, LIN7C, KPNA6, and RAB11A, are regulated by four piRNAs (*piR-38240*, *piR-34393*, *piR-40666*, and *piR-51810*), which could lead to abnormal cell death, cellular homeostasis, and the transport of Aβ in the AD process [145]. In the human brain, 81 up-regulated and 22 down-regulated piRNAs were detected, of which 24 were specifically expressed [144]. Additionally, these AD-associated piRNAs were strongly correlated with AD-risk DNA variants (SNPs) through eQTL analysis [144]. Further research found associations of 17 variants across 8 loci, including *APOE*, *APOJ*, and *SLC24A4*, which were genome-wide significant [146]. Similarly, Mao, Fan et al. detected 9453 piRNAs from 4 AD patients, of which *DQ570746* and *DQ582201* accelerate aging by down-regulating *CYP19A1* and *CD33* in a TE-independent manner [147]. 

PiRNAs were first reported in the human cerebrospinal fluid exosomes of AD dementia patients [148]. Among them, down-regulated *piR-019324* and up-regulated *piR-019949/-020364* were identified in the exosomes, which may modulate the tau and Aβ42/40 in mild cognitive impairment [148]. Additionally, in AD plasma-derived extracellular vesicles, up-regulated piR-000552 and -020450 were identified [149]. These piRNAs may serve as candidate non-invasive biomarkers to distinguish between clinical and preclinical AD. However, the sensitivity and specificity of exosomal piRNAs still need to be improved.

Currently, three authors have summarized the potential role of piRNAs in AD [24,137,150]. Therefore, extensive studies should focus on seeking novel piRNAs and their targets, especially in lncRNAs and snRNAs. Additionally, the application of GWAS in piRNA research will greatly advance the understanding of Alzheimer’s disease pathology.

### 5.2. Parkinson’s Disease (PD)

PD is the second most common neurodegenerative disease, which is characterized by the gradual reduction of dopaminergic neurons in the substantia nigra pars compacta [151,152]. The heterochromatin decompensation and reduction of piRNAs/PIWI drive transposable element dysregulation in tauopathy, which is one of the pathological changes in PD [27,153].

#### 5.2.1. PD Models

In a *C. elegans* model of PD, 79 up- (most significant: *21ur-5499*) and 33 down-regulated (most significant: *21ur-10292*) piRNAs have been found [116]. Among them, *21ur-1412* may target two PD-relevant genes (*T08G11.1* and *rab-39*), while *21ur-10487, -839*, and *-2781* may target four PD-relevant genes (*tpa-10, taf-1, dop-3*, and *rem-8*) [116]. These targets act as effective signatures for PD: dysregulated *Tpa-1* causes metabolic imbalances in a-synuclein [154]; defects in *T08G11.1* that are related to membrane lipid homeostasis appear in VPS13C-PD cases [155]; the absence of *taf-1* (TAFA in humans), *dop-3* (DRD2 in humans), and *rme-8* (DNAJC13 in humans) leads to dopaminergic dysfunction in PD brains [156,157,158]. In two other *C. elegans* PD models (α-syn(A53T)^Tg^ and Aβ^Tg^; α-syn(A53T)^Tg^ transgenic strains), *21ur-10824, 21ur-11898*, and 21ur-13215 were up-regulated [28], which is consistent with previous studies of PD [159]. What is more, in these two strains, the deletion of *tofu-1* (a piRNA biogenesis gene) ameliorates behavioral phenotypes, improves thrashing, extends the lifespan, reduces α-synuclein expression, and alleviates dopaminergic neuron degeneration [28]. These results underscore the indispensable roles of neuronal piRNAs in the molecular basis of PD.

#### 5.2.2. PD Patients

In PD patient studies, many dysregulated piRNAs were detected in fibroblasts (113 up- and 168 down-regulated), IPSCs/ESCs (193 up- and 62 down-regulated), and neurons (150 up- and 329 down-regulated) [159]. Among these, the fraction of memory piRNAs in neurons was deregulated and predominantly enriched in both SINE and LINE [159]. In one study of PD patients, 20 piRNAs were significantly altered in the prefrontal cortex, with 55 altered piRNAs in the amygdala, and *hsa-piR-748391* was co-expressed in both regions [25]. In addition, *hsa-piR-131693* levels decreased as the PD progressed. In the prefrontal cortex, transposon-derived piRNAs were massively detected [25], and their targets (pseudogenes *HSP90AA1* and *EEF1A1*) have been reported to regulate the abnormal aggregation of α-synuclein and neuronal damage [160,161]. Because multiple piRNAs have a broad targeting capacity, the above piRNAs still need further verification or refinement.

The accumulating evidence suggests that piRNAs are critical regulators affecting the etiology of PD [8,110,162]. Future work should focus on exploring comprehensive piRNA databases across different species and cell types, as well as on identifying specific temporal and spatial expression patterns in tissues.

### 5.3. Amyotrophic Lateral Sclerosis (ALS)

ALS is a progressive motor neurodegenerative disease that causes selective motor neuron loss, progressive muscle wasting, and, ultimately, death [163]. Dysregulated TDP-43 and FUS proteins contribute to ALS pathogenesis by causing aberrant post-translational modification and subcellular mis-localization in neurons [163,164,165,166]. Recent research has highlighted the critical roles of piRNAs/PIWI in ALS [45].

#### 5.3.1. ALS Models

After specifically knocking down the cabeza (Caz, a FUS homolog) in *Drosophila,* Aub was overexpressed, promoting crawling speed and climbing ability in both larvae and adults [118]. In the larvae CNS, knocking down Caz increased the level of pre-piRNAs and co-immunoprecipitation in the nuclears, while Aub overexpression had the opposite effect [118]. In addition, Caz knockdown decreased mature-side piRNA expression, while Aub overexpression or inhibition had no effect [118]. Overall, these results suggest that Caz negatively regulates the expression levels of pre-piRNAs and mature piRNAs, contributing to ALS. 

#### 5.3.2. ALS Patients

In ALS human brains, three down-regulated piRNAs (*hsa-piR-000578, -020871*, and *-002184*) and two up-regulated piRNAs *(hsa-piR-009294* and *-016735*) were identified [133,167]. Meanwhile, protein-level analysis also showed a decrease in PIWIL1 and an increase in PIWIL4 [167]. These dysfunctional PIWIL1-directed piRNAs mislocate with TDP-43 in cytoplasmic inclusions, which may be an important determinant of TDP-43 accumulation in the cytoplasm and may contribute to ALS pathogenesis [167]. In the biofluids of ALS, up-regulated piRNAs in both CSF (*hsa*-*piR-020326*, *-020365*, and *-006890*) and serum (*hsa*-*piR-006890*, *-008114*, *-000775*, and *-000765*) have been identified by RNA-seq [168]. In addition, as diagnostic biomarkers, serum piRNAs may have more diagnostic value compared to cerebrospinal fluid (CSF). A comprehensive assessment of cell-free piRNAs/PIWI complexes in the context of ALS diagnosis is still needed.

Currently, the study of novel piRNAs in ALS relies on high-throughput, long-read, and extensive platforms. Simultaneously, their biological functions still warrant further clarification through a functional analysis of ALS pathophysiology.

## 6. Conclusions

Although many pieces of evidence indicate the fundamental function of piRNAs in diverse neural activities and their association with disease occurrence, elucidating the molecular mechanisms underlying neuronal piRNA-mediated gene regulation is just beginning. PiRNA research still faces major limitations, such as non-obvious targets, imperfect databases, multiple genomic locations, and lower abundance. Several neurodegenerative diseases have shown increased transposon expression, yet the roles of promoter and TE piRNAs in regulating neural development and degenerative diseases still need to be explored.

## Figures and Tables

**Figure 1 genes-15-00653-f001:**
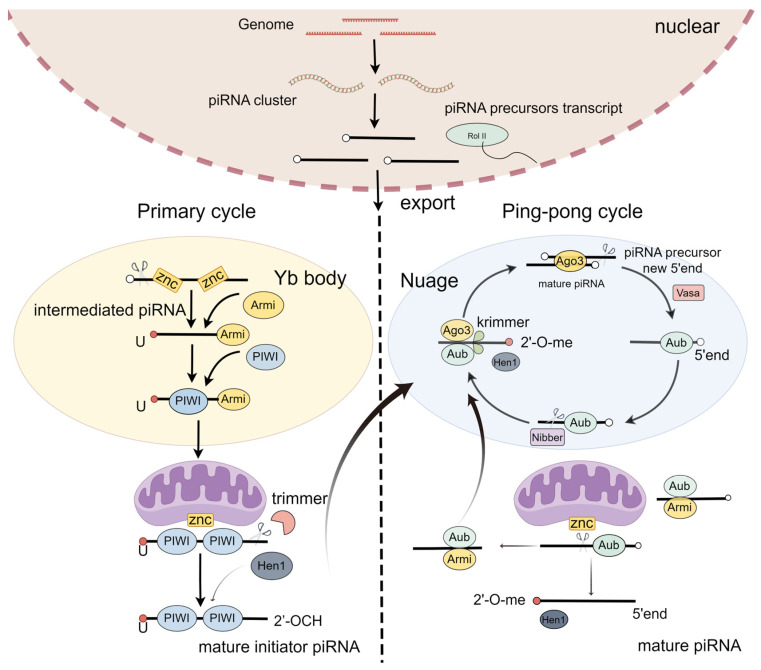
PiRNA biogenesis.

**Figure 2 genes-15-00653-f002:**
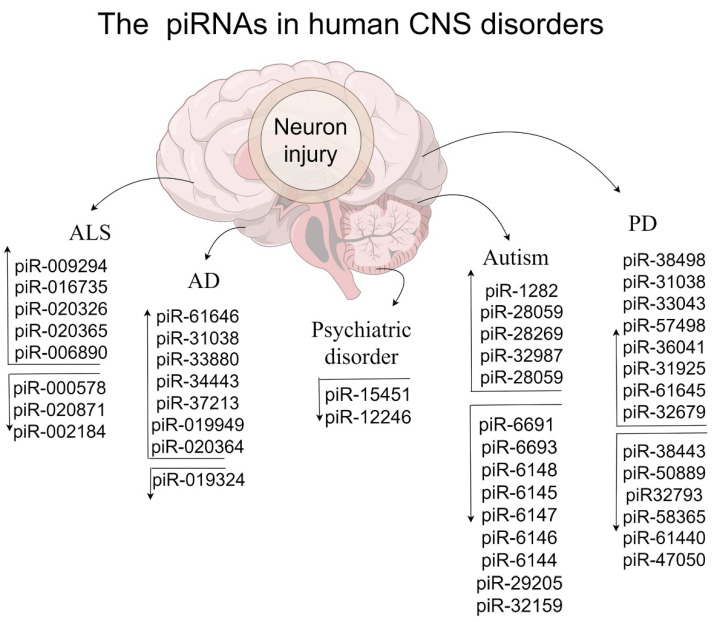
The influence of human piRNAs in CNS disorders.

**Table 1 genes-15-00653-t001:** The relevant piRNA databases.

Database	Species	Function	Website	Reference
Public databases				
NONCODE	Covers 17 species (Human, Mouse, Cow, Rat, Pig, *Drosophila*, and Nematode)	annotation, potential, function, and mapping	http://www.noncode.org/, accessed on 9 May 2024	[72,73]
deepBase	Insect, Nematode, Insect, and Mammal	expression profile and interaction function	http://rna.sysu.edu.cn/deepbase3/index.html, accessed on 9 May 2024	[74,75]
Specific databases				
PiRbase	Covers 44 species (Human, Mouse, Rat, Zebrafish, Chicken, Silkworm and Rabbit)	new sequence, gene target, gold standard set, cluster, function, and network	http://bigdata.ibp.ac.cn/piRBase, accessed on 9 May 2024	[76]
piRNA bank	Human, Mouse, and Rat	piRNAs cluster, map, and homologous and chromosomal locations	http://pirnabank.ibab.ac.in/, accessed on 9 May 2024	[77]
PiRNAclustersDB	Cover 51 species (Mollusks, Arthropods, Fish, Amphibians, Reptiles, and Mammals)	piRNAs cluster database	https://www.smallrnagroup.uni-mainz.de/piRNAclusterDB, accessed on 9 May 2024	[78]
piRTarBase	*C. elegans*	target prediction	http://cosbi6.ee.ncku.edu.tw/piRTarBase, accessed on 9 May 2024	[79]
piRNA-IPdb	Mouse	length distribution, nucleotide composition, and expression level	https://ipdb2.shinyapps.io/ipdb2/, accessed on 9 May 2024	[80]

**Table 2 genes-15-00653-t002:** The piRNAs in CNS disorder models.

Disease	Species	Models	Results	Reference
stroke	Rat	ischemia	54 up- and 51 down-regulated piRNAs were detected	[114]
		ischemia tolerance	*piR-000618*, *-017990*, and *-014971* are up-regulated by targeting CREB2	[89]
Gulf War illness	Rat	GWI-relevant exposures	*piR-007899* and *piR-019162*	[115]
nerve injury	Mice	MIWI/PIWIL1 KO	up-regulated *piR-14384*, *-69959*, *-52953*, and *-52274*	[19]
AD	*C. elegans*	Overexpressing α-syn pan neuron	*21ur-10488* and *21ur-2781* were related to the AD pathology	[116]
	*Drosophila*	tau transgenic	down-regulated Ago3	[117]
		Ago3/PIWIL1 KD in tau transgenic	improvement of locomotor defect	[27]
PD	*C. elegans*	tau transgenic	79 up- and 33 down-regulated piRNAs	[117]
		Overexpressing A53T pan neuron and co-expressing Aβ and α-syn pan neuron	up-regulated *21ur-10824*, *21ur-11898*, and *21ur-13215*	[28]
ALS	*Drosophila*	Caz KD	Aub was overexpressed, while pre-piRNAs were increased	[118]

## Data Availability

No new data were created or analyzed in this study. Data sharing is not applicable to this article.

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
