# Peer review of "PIWI-Interacting RNAs: A Pivotal Regulator in Neurological Development and Disease"

_genes, 2024, doi:10.3390/genes15060653_

Round 1
Reviewer 1 Report
Comments and Suggestions for Authors
The review by Pan and colleagues collects and summarizes a large number of publications that contain a great deal of information related to piRNAs in nervous tissues. Some comments are provided below.
The title is confusing. The title suggests as main subjects, pathways and piRNA molecules, while it seems the manuscript focuses on the latter. The second part of the title suggests the manuscript will focus on neuronal development and related disorders. Yet ALS, Alzheimer’s and Parkinson’s diseases for example are age-related neuronal disorders and not developmental disorders to my knowledge.
The manuscript overall lacks depth and clarity. Most of the text simply summarizes the main findings of published reports without an evaluation of the value of the report. For example, it lists the up and down regulated piRNAs genes found in a particular neuron related disease. Were any of these piRNAs or their targets validated as actually being involved in the disease ? What did the authors find beyond the abstract that was an advance in the understanding. Too often, authors state something like “However,
the precise mechanism is still obscured. To further enhance our understanding, we summarize …….” The authors then review the publications cited in a few brief sentences without further analysis. Without additional depth and clarity on a reading of the reviewed manuscripts, the reader is left with the same result as reading a list of abstracts on the topic. There is too much cataloging of piRNAs without providing depth such as: 1) what additional experiments were conducted other than xxx piRNAs were upregulated and xxx piRNAs were downregulated; 2) how did this publication advance the field in terms of understanding and mechanism; 3) what specific new knowledge to the field as a whole did it demonstrate; 3) what new directions did the publication provide or point other scientists to explore ? 4)what existing gaps in knowledge did the publication fill.
There was little new synthesis of knowledge based upon a review of the literature. What was found across studies, for example: 1) what trends were observed from reading across multiple studies and publications ? 2) what overlaps, if any were found 3) what emerged from reading the large number of publications (e.g. confirmations based on different perspectives, controversies or disagreements), that is not apparent from reading individual publications.
The Elephant in the room: Biosynthesis and basic biology of piRNAs is based mostly upon findings from the development field, yet the review is about tissues also outside germline cells. The review does not make a distinction in biosynthesis and molecular function of piRNAs from germline or nervous tissue. Two major questions: 1) how are piRNAs synthesized and amplified in non-germline cells such as nervous tissue, and how do they contribute to gene regulation, if via transposon silencing, then where are the argonautes in nervous tissue and what is the status of transposons. If via silencing of protein coding genes, how does this happen in nervous tissue?
The review of the major piRNA databases lacks information on whether these are actually working or accessible now. How can they be “relevant” as authors state or important if they don’t work or are not currently supported ?
Comments on the Quality of English LanguageRequires English language revision.
Author Response
COMMENTS TO THE AUTHOR:
Reviewer #1
The review by Pan and colleagues collects and summarizes a large number of publications that contain a great deal of information related to piRNAs in nervous tissues. Some comments are provided below.
We thank the reviewer for providing instructive comments to this manuscript. We have revised the manuscript according to reviewers’ suggestion and significantly improved the manuscript.
Major comment
1) The title is confusing. The title suggests as main subjects, pathways and piRNA molecules, while it seems the manuscript focuses on the latter. The second part of the title suggests the manuscript will focus on neuronal development and related disorders
We are grateful for the reviewer’s helpful comments and suggestions. Our manuscript mainly focuses on describing the relationship between piRNA and its pathways in CNS diseases, and some neurobehaviors factors such as food intake, memory and leaning. What’s more, we also classify ALS, AD, and PD as neurodegenerative diseases. In this revision, we have changed our title as “PIWI-Interacting RNAs Pathway: A Novel Regulator in Central Neural Development and Degenerative Disorders”, and revised the mentioned description in Abstract (Line 17) and Introduction (Line 40, 49).
2) The manuscript overall lacks depth and clarity. Most of the text simply summarizes the main findings of published reports without an evaluation of the value of the report.
We are grateful for the reviewer’s helpful comments and suggestions. Although mRNA and ncRNA therapy have great potential for the treatment of neuronal diseases, only the identities of piRNA remain to be explored. Most of research only focused on the sequencing. In this revision, we have reviewed and summarized the insightful findings, particularly the role of piRNA in neurodevelopmental and neurodegenerative disease. We have changed the mention descriptions in (Line 135-137, 155-156, 168-170, 185-187, 210, 226-227, 241-244, 275-278, 299-305, 311-313, 324-328, 332- 338, 343-345, 354-357, 384-387, 397-402, 403-406, 420-426, 437-439, 440-443, 463-465, 467-473) and conclusion (line 478-481)
3) The Elephant in the room: Biosynthesis and basic biology of piRNAs is based mostly upon findings from the development field, yet the review is about tissues also outside germline cells. The review does not make a distinction in biosynthesis and molecular function of piRNAs from germline or nervous tissue. Two major questions: 1) how are piRNAs synthesized and amplified in non-germline cells such as nervous tissue, and how do they contribute to gene regulation, if via transposon silencing, then where are the argonauts in nervous tissue and what is the status of transposons. If via silencing of protein coding genes, how does this happen in nervous tissue?
We like the reviewer’s idea. Unfortunately, current research on piRNA synthesis and amplification is mainly focused on reproductive systems. The specific mechanism of piRNAs in brain have not been clearly determined[1-3]. Most article only focus on determining the quantity or identify of piRNA[4-6]. But your suggesting has put forward a very important direction for this field. According to your suggesting, we have been added relevant summary in (Line 275-278 and Line 478-479).
4) The review of the major piRNA databases lacks information on whether these are working or accessible now. How can they be “relevant” as authors state or important if they don’t work or are not currently supported?
We appreciate the constructive question raised by the reviewer. We have carefully rechecked the availability of these databases and remove the invalid website. Meanwhile, we have increased the scope and function of application of these databases in Table1.
References:
- Lingjun Zuo1, Zhiren Wang2, Yunlong Tan2, Xiangning Chen3, and Xingguang Luo1,2,*. piRNAs and Their Functions in the Brain. HHS Public Access 2016.
- Kim, K.W. PIWI Proteins and piRNAs in the Nervous System. Mol Cells 2019, 42, 828-835, doi:10.14348/molcells.2019.0241.
- Lee, E.J.; Banerjee, S.; Zhou, H.; Jammalamadaka, A.; Arcila, M.; Manjunath, B.S.; Kosik, K.S. Identification of piRNAs in the central nervous system. RNA 2011, 17, 1090-1099, doi:10.1261/rna.2565011.
- Beversdorf, D.Q.; Sohl, K.; Levitskiy, D.; Tennant, P.; Goin-Kochel, R.P.; Shaffer, R.C.; Confair, A.; Middleton, F.A.; Hicks, S.D. Saliva RNA Biomarkers of Gastrointestinal Dysfunction in Children With Autism and Neurodevelopmental Disorders: Potential Implications for Precision Medicine. Front Psychiatry 2021, 12, 824933, doi:10.3389/fpsyt.2021.824933.
- Roy, J.; Sarkar, A.; Parida, S.; Ghosh, Z.; Mallick, B. Small RNA sequencing revealed dysregulated piRNAs in Alzheimer's disease and their probable role in pathogenesis. Mol Biosyst 2017, 13, 565-576, doi:10.1039/c6mb00699j.
- Zhang, T.; Wong, G. Dysregulation of Human Somatic piRNA Expression in Parkinson's Disease Subtypes and Stages. Int J Mol Sci 2022, 23, doi:10.3390/ijms23052469.
Reviewer 2 Report
Comments and Suggestions for Authors
This appears to be an interesting, albeit difficult to comprehend due to poor language communication, review describing the potential role of piwiRNAs in the etiology of several neurological and psychiatric disorders. Two major suggestions to the authors are:
1. please provide greater details of the prior piwiRNA research. For example, while the authors have provided information, whether they describe animal or human-based studies for some of the cited reports, most of the cited studies have no such information. This is important, especially in the context of studying psychiatric disorders, which are uniquely human and difficult to recapitulate in animal models.
2. It would be interesting if the authors could cross-reference some of the cited piwiRNA expression studies, either in the brain or other tissues, with GWAS studies. Given that GWAS are now conducted on much larger cohorts than any existing expression data, any supporting evidence from the GWAS could reinforce the piwiRNAs as potential etiological factors in the development of psychiatric disorders.
Comments on the Quality of English LanguageNumerous grammatical and semantic errors. I would encourage the authors to proofread their manuscript before sending it out.
Author Response
Reviewer #2 This appears to be an interesting, albeit difficult to comprehend due to poor language communication, review describing the potential role of piwiRNAs in the etiology of several neurological and psychiatric disorders.
We are grateful for the reviewer’s helpful comments and suggestions. We have carefully checked the whole manuscript.
Major comment:
1) please provide greater details of the prior piwiRNA research. For example, while the authors have provided information, whether they describe animal or human-based studies for some of the cited reports, most of the cited studies have no such information. This is important, especially in the context of studying psychiatric disorders, which are uniquely human and difficult to recapitulate in animal models.
The reviewer has pointed out very good questions. Unlike the miRNAs, the precise mechanism of piRNAs in CNS is still unclear. In this revision, we have provided more details of the previous piRNA research (Line 299-305, 324-326, 343-345, 377-380, 384-389, 397-400, 420-426, 463-465), and carefully checked the citations to ensure their accuracy and provide readers with a clear source of evidence.
2)It would be interesting if the authors could cross-reference some of the cited piwiRNA expression studies, either in the brain or other tissues, with GWAS studies. Given that GWAS are now conducted on much larger cohorts than any existing expression data, any supporting evidence from the GWAS could reinforce the piwiRNAs as potential etiological factors in the development of psychiatric disorders.
We like the reviewer’s idea. We have carefully search the article that closely related to piRNA and GWAS in the nervous system, and only 2 research have been found (PMID: 28654860, PMID: 28770390). In this revision, we have added relevant content (Line 389-492) and mentioned the corresponding issues (Line 405-406) in the summary paragraph, and your suggestions will provide a reliable suggesting for the filed.
Reviewer 3 Report
Comments and Suggestions for Authors
In this manuscript entitled "PIWI-Interacting RNAs Pathway: A Novel Regulator in Central Neural Development and Related Disorders”, the authors reviewed the biological characteristics and functional roles of piRNAs in the neural development and neurocognitive behaviors. In addition, this paper highlights recent studies on the investigation of the roles of piRNAs in several CNS diseases, including neurodevelopmental and neurodegenerative diseases. The authors also suggested that the functional significance of piRNAs in future RNA-based therapeutics for CNS disorders. Following are my concerns that the authors should address to improve the paper.
Major comments
1. I recommend the authors cite following relevant and essential papers in their manuscript. Add the following citations to the sentence describing the detail of neurocognitive behaviors (Craske et al., 2009, PMID: 19957279; Cooke et al., 2022, PMID: 35530177). In addition, add a citation to the sentence discussing the relationship between RNA homeostasis and neural disorders (Kwon et al., 2023, PMID: 37968980) as well as emerging interest in RNA Therapeutics for genetic diseases (Caroll, 2023, PMID: 36482771).
Minor comments
1. Multiple typos and grammatical errors occurred throughout the manuscript. Following are some examples.
1) In Table 1, please remove the hyperlink in the table.
2) In Table 2, please correct ‘C.elecan’ to ‘C. elegans’.
3) On page 6, line 218, please correct “3.3.3” to “3.3.2”. Correspondingly, the section “3.3.4. Food Intake” should be “3.3.3 Food Intake”.
2. It will be better to enhance the resolution of the figures.
3. Please italicize “C. elegans” written throughout the manuscript.
Comments on the Quality of English LanguageIt is fine.
Author Response
Reviewer #3: In this manuscript entitled "PIWI-Interacting RNAs Pathway: A Novel Regulator in Central Neural Development and Related Disorders”, the authors reviewed the biological characteristics and functional roles of piRNAs in the neural development and neurocognitive behaviors. In addition, this paper highlights recent studies on the investigation of the roles of piRNAs in several CNS diseases, including neurodevelopmental and neurodegenerative diseases. The authors also suggested that the functional significance of piRNAs in future RNA-based therapeutics for CNS disorders. Following are my concerns that the authors should address to improve the paper.
We thank the reviewer for providing instructive comments to this manuscript. We have revised the manuscript according to the reviewer’s suggestions and significantly improved the manuscript.
Critical comments:
The manuscription should cite Craske et al., 2009ï¼›PMID: 19957279; Cooke et al., 2022, PMID: 35530177; Kwon et al., 2023, PMID: 37968980ï¼›Caroll, 2023, PMID: 36482771
We are grateful of the reviewer for recommending several articles to the manuscript, making our manuscript more comprehensive. These important articles have been cited in the appropriate paragraphs (line 231- 232, 280-281, 332-333 and 359-361) and corresponding references.
Mini comments
Multiple typos and grammatical errors occurred throughout the manuscript
1) In Table 1, please remove the hyperlink in the table.
We have removed the hyperlink in Table 1
2) In Table 2, please correct ‘C.elecan’ to ‘C. elegans’.
In the table 2, we have substituted ‘C.elecan’ to ‘C. elegans’.
3) On page 6, line 218, please correct “3.3.3” to “3.3.2”. Correspondingly, the section “3.3.4. Food Intake” should be “3.3.3 Food Intake”.
We have changed the wrong subtitles (Line 225, and Line 239)
- It will be better to enhance the resolution of the figures.
We have changed the resolution of the figures in Line 59-61 and Line 287-289
- Please italicize “C. elegans” written throughout the manuscript.
We have changed all the Latin species to italics throughout the manuscript.
Reviewer 4 Report
Comments and Suggestions for Authors
The review of PIWI RNAs (piRNAs) provides a thorough summary of the roles that these molecules play in several biological processes, most notably those that affect the central nervous system (CNS). Nonetheless, a few recommendations may improve the review's accuracy and clarity. Although the section on piRNA databases offers a comprehensive overview, it is unclear what each database's particular purposes and scope are. This section would be more instructive if there were brief explanations or instances of how they may be used.
The review covers the dysregulation of piRNAs in several disorders, however it would be helpful to go into further detail on how this dysregulation affects function. What possible therapeutic approaches could target these dysregulations and how do they contribute to the pathophysiology of disease?
Given the intricacy of piRNA research, it would be beneficial to the review's credibility to acknowledge methodological constraints, such as biases in sequencing techniques or difficulties in target prediction, since these would give context for evaluating the results.
The review might be concluded more satisfactorily with a fuller synopsis of the major discoveries and a discussion of potential directions for future study, especially in the context of CNS illnesses.
Author Response
Reviewer #4: The review of PIWI RNAs (piRNAs) provides a thorough summary of the roles that these molecules play in several biological processes, most notably those that affect the central nervous system (CNS).
1) Although the section on piRNA databases offers a comprehensive overview, it is unclear what each database's particular purposes and scope are. This section would be more instructive if there were brief explanations or instances of how they may be used.
We are pleased with the comments provided by the reviewer. To make table clearer, we have added the species range and applicable functions of piRNAs in Table1.
2) The review covers the dysregulation of piRNAs in several disorders, however it would be helpful to go into further detail on how this dysregulation affects function. What possible therapeutic approaches could target these dysregulations and how do they contribute to the pathophysiology of disease?
We like the reviewer’s idea. Although mRNA and ncRNA therapy have great potential for the treatment of neuronal diseases, only the identities of piRNA remain to be explored. Such as in Parkinson disease, the heterochromatin decompensation and reduction of piRNAs/PIWI drive transposable element dysregulation in tauopathy, which is one of the PD pathological changes. In this revision, we have added the mentioned descriptions and further summarized this constructive question, especially in the areas of neurodevelopmental and neurodegenerative diseases. We have changed the mention descriptions in (Line 135-137, 155-156, 168-170, 185-187, 210, 226-227, 241-244, 275-278, 299-305, 311-313, 324-328, 332- 338, 343-345, 354-357, 384-387, 397-402, 403-406, 420-426, 437-439, 440-443, 463-465, 467-473) and conclusion (line 478-481).
3) Given the intricacy of piRNA research, it would be beneficial to the review's credibility to acknowledge methodological constraints, such as biases in sequencing techniques or difficulties in target prediction, since these would give context for evaluating the results. The review might be concluded more satisfactorily with a fuller synopsis of the major discoveries and a discussion of potential directions for future study, especially in the context of CNS illnesses.
The piRNAs research still persists its major limitations, such as nonobvious targets, im-perfect databases, multiple genomic locations and lower abundance. The promoter and TE piRNAs will be further identified in CNS disorders. We have changed the mentioned description in conclusion (Line 478-482).
Round 2
Reviewer 1 Report
Comments and Suggestions for Authors
Authors have provided satisfactory responses to my queries.
Comments on the Quality of English LanguageSome corrections should be made to the English language style. The newly added text especially needs attention.
Author Response
We are sorry for the grammar mistakes. We have carefully checked throughout the manuscript.
Reviewer 2 Report
Comments and Suggestions for Authors
While the revised manuscript has improved by providing additional information, the thematic organization of the manuscript is somewhat disjointed. For example, it is unclear what the authors mean by saying psychiatric disorders. ASD is considered a psychiatric disorder but is discussed separately from "psychiatric disorders".
Additionally, the authors also discuss the piRNAs' role in food intake, but little can be gleaned from the section. It is unclear whether the reported results describe a disease-related pathology, or normal fluctuations in the metabolic state that may promote an increase in weight.
Comments on the Quality of English LanguageThe English language of the revised manuscript still needs to be improved, but the revised manuscript reads better than the first submission.
Author Response
Taking the reviewer's constructive suggestion into account, we have moved the ASD section to part 4.2.2 and renamed section 4.2.1 as SCZ.
Food intake, a complex physiological process, is regulated by the CNS through neuropeptides and neurotransmitters. PiRNAs in the CNS have been highlighted for their potential. However, only a few studies have been conducted on locusts. In this revision, we have updated the mentioned descriptions and added the related references.
We apologize for the grammatical mistakes. We have carefully checked the entire manuscript and made it more readable.
Reviewer 4 Report
Comments and Suggestions for Authors
The authors made the requested changes by implementing the text.
Author Response
We apologize for the grammatical mistakes. We have carefully checked the entire manuscript.